# Macro and Micro-Nutrient Accumulation and Partitioning in Soybean Affected by Water and Nitrogen Supply

**DOI:** 10.3390/plants12091898

**Published:** 2023-05-06

**Authors:** Ingrid Silva Setubal, Aderson Soares de Andrade Júnior, Silvestre Paulino da Silva, Artenisa Cerqueira Rodrigues, Aurenívia Bonifácio, Evandro Henrique Figueiredo Moura da Silva, Paulo Fernando de Melo Jorge Vieira, Rafael de Souza Miranda, Nicolas Cafaro La Menza, Henrique Antunes de Souza

**Affiliations:** 1Postgraduate Program in Agronomy, Agricultural Science Center, Federal University of Piauí, Teresina 64048-550, Brazil; ingrid.s.setubal@gmail.com (I.S.S.); silvestre.mapito@gmail.com (S.P.d.S.); artenisacerqueira@ufpi.edu.br (A.C.R.); 2Embrapa Mid North, Brazilian Agricultural Research Corporation, Teresina 64008-780, Brazil; aderson.andrade@embrapa.br (A.S.d.A.J.); paulofernando.vieira@embrapa.br (P.F.d.M.J.V.);; 3Department of Biology, Center of Nature Science, Federal University of Piauí, Teresina 64600-000, Brazil; aurenivia@ufpi.edu.br; 4Luiz de Queiroz College of Agriculture, University of São Paulo, Piracicaba 13418-900, Brazil; ehfmsilva@gmail.com; 5Campus Profa. Cinobelina Elvas, Federal University of Piauí, Bom Jesus 64900-000, Brazil; 6West Central Research, Extension and Education Center, University of Nebraska-Lincoln, Lincoln, NE 68588, USA; ncafarolamenza2@unl.edu

**Keywords:** *Glycine max*, biomass, plant nutrition, water stress, soybean yield, nitrogen, symbiotic fixation

## Abstract

This study aimed to investigate the influence of water availability and nitrogen fertilization on plant growth, nutrient dynamics, and variables related to soybean crop yield. Trials were performed in Teresina, Piauí, Brazil, using randomized blocks in a split-split plot arrangement. The plots corresponded to water regimes (full and deficient), the split plots to N fertilization (0 and 1000 kg ha^−1^ N-urea), and the split-split plots to harvest times of soybean plants (16, 23, 30, 37, 44, 58, 65, 79 and 86 days after emergence), with three replicates. In general, the accumulation and partitioning of nitrogen (N), phosphorus (P), potassium (K), calcium (Ca), magnesium (Mg), sulphur (S), copper (Cu), iron (Fe), manganese (Mn), zinc (Zn) and boron (B) were decreased in plants subjected to water deficit and without N fertilization. Although nitrogen fertilization promoted elevated N accumulation in tissues, it did not result in any significant yield gain, and the highest seed yields were found in plants under full irrigation, regardless of N supplementation. However, deficient irrigation decreased the seed oil content of N-fertilized plants. In conclusion, N fertilization is critical for nutrient homeostasis, and water availability impairs biomass and nutrient accumulation, thereby limiting soybean yield performance.

## 1. Introduction

Modern soybean cultivars produce seeds with a high content of proteins and essential amino acids for humans and animal feed, and are a key component in industrial raw materials [1]. In the 2021–2022 season, 357.9 million Mg of soybeans were produced worldwide, and Brazil was the world’s largest producer, with 129.5 million Mg [2]. Soybean is a crop with a high demand for nitrogen (N), a nutrient that plays a central role as a structural component of amino acids, proteins, and chlorophylls. It also comprises other N-containing compounds, such as hormones and nucleic acids, thereby playing a critical role in cell function and plant metabolism [3].

Nearly 80 kg of N is necessary to produce 1.0 Mg of soybean seeds as 65 kg of N is exported to the seeds [4]; therefore, approximately 480 to 640 kg N ha^−1^ is needed to attain yields between 6.0 and 8.0 Mg ha^−1^ [5]. The data clearly demonstrate that high N levels are critical for proper soybean production. In contrast, nitrogen deficiency affects the physiological functioning and metabolic activities of soybean plants, including inhibition of leaf growth, reduction in photosynthetic rate, and changes in the volume and length of the root system [6].

In agricultural systems, the N requirements of soybean plants have been met through the biological fixation of atmospheric N (BNF). BNF is a process of capturing atmospheric N by diazotrophic microorganisms that establish an efficient symbiotic relationship with the crop and provide N in the form of ureides [7]. In Brazil, inoculants are an indispensable technology for soybean cultivation and constitute one of the most relevant factors for high productivity as an economical method comparable to nitrogen fertilizers. On the other hand, the application of N as topdressing on soybean fields is a common practice in the United States and China because the sole use of BNF does not meet the N demand of modern soybean cultivars and therefore limits crop productivity [8].

Knowledge of nutrient uptake, accumulation, and partitioning curves in soybean plants at different development stages is key to determining the periods of greatest nutritional demand and, thus, performing a more precise nutrient application [9,10,11,12,13]. In addition, some studies have reported controversial data about whether BNF can meet the N demand of soybeans grown in major producing areas [14,15], and there is no information concerning the level and production circumstances under which BNF could be insufficient to meet the crop N demand. Nutritional requirements may vary according to crop management and specific characteristics, such as cultivar/variety, growth habit, and cultivation cycle [12]. Additionally, irrigation has been shown to be an effective practice in improving crop yields and seed quality since the availability of nutrients in soil is directly related to an adequate water supply [13]. Soil moisture has an influence on both the yield and seed quality attributes of soybean. In evaluating the effects of soil moisture during the reproductive phase of soybean plants, studies have shown that the maintenance of optimal soil moisture content during the reproductive phase is important for increased accumulation of protein, sucrose, oil, and macronutrients in the seeds [14,16].

In rainfed farming, sowing must be carried out at the beginning of the rainy season to ensure adequate water availability and to avoid the negative influence of water deficiency on the plants’ metabolic processes. Drought is a major threat to crop production worldwide, and legumes are sensitive to drought stress by negatively affecting BNF, grain yield and protein content in seeds [17]; however, some results have shown that a rate of 50 kg N ha^−1^ does not impair BNF and can deliver 71% of total N uptake [18]. This is especially relevant in Brazil’s agricultural frontiers, such as the specific savanna region (Cerrado) in northeastern Brazil, where there is a lack of site-specific research regarding the use of inoculants and nutrient cycling of organic matter in annual crops. In addition, the prevailing local edaphoclimatic conditions (soils with low clay contents and high temperatures) increase the challenge of adopting sustainable agricultural practices.

Here, we hypothesize that (i) N application could result in nutrient accumulation and yield gains in soybean plants under drought stress at the reproductive stage and (ii) the combination of N fertilization and the water regime modifies nutrient uptake and partitioning by soybean plants. This hypothesis was tested by employing trials with soybean plants subjected to different levels of water availability and N fertilization and combining analyses of growth with biochemical and yield components.

## 2. Results

### 2.1. Water Regimes and Soil Moisture

At the initial stage of the experiment (0 to 35 DAE), before differentiating the water regimes (WR), irrigation depths applied to strips 1 and 2 were equivalent to 125.3 and 127.1 mm (where, later, WRs at 50 and 100% of ETc were applied, respectively), demonstrating that the WRs were similar in the experimental area. During the entire crop cycle, the total accumulated irrigation depths applied to soybean plants were 345.8 and 465.2 mm for the deficient and full WR treatments, respectively (Figure 1).

Soil moisture levels varied according to the application of WRs, especially in the differentiating phase (water stress period). Soil moisture was maintained above the critical moisture content in the 0.0–0.3 m and 0.3–0.6 m layers in the area cultivated with full WR. Even after the end of irrigation application, the area irrigated with 50% ETc retained low soil moisture levels (Figure 1B,C).

### 2.2. Dry Biomass Accumulation

Dry biomass (DM) accumulation was altered in response to different WR and N supplementation (Table 1). The maximum DM accumulation was considerably higher in soybean plants cultivated under full WR compared to deficient WR treatment. The differences between DM production under full and deficient WR were more evident at the R4 stage (44 DAE) (Figure 2). N supplementation provided punctual differences in total DM at V9 (30 DAE) and R6 (65 DAE) in plants cultivated under full WR, in which the accumulation of DM was lower without N supplementation. Lower DM accumulation occurred at R1 (37 DAE) and R3 (44 DAE) in soybean cultivated under deficient WR without N fertilization. The DM of soybean leaves under both WRs and stem DM in plants under full WR and supplemented with N were higher than those recorded in plants cultivated without N fertilization during physiological maturity.

The maximum leaf DM accumulation was recorded at the R5 stage (60 DAE) in N-fertilized plants under full WR, while in the other treatments, the maximum DM in leaves was recorded at R4 (54 DAE). On average, deficient WR reduced stem DM by 65% and 56% in plants with and without N supplementation, respectively, compared to that of plants under full WR. There was a sharp drop in DM when plants reached physiological maturity, with stem DM contributing just over 20% to the total DM of the plant (Appendix A).

The maximum estimated accumulation of total DM in soybean plants cultivated in full WR with and without N fertilization was 11,548 kg ha^−^^1^ and 10,152 kg ha^−^^1^, respectively. In plants under deficient WR, a total of 6457 and 4501 kg ha^−^^1^ of maximum DM accumulation was recorded with and without N fertilization, respectively (Appendix A).

### 2.3. Macronutrient Accumulation

The differentiation of WR influenced macronutrient accumulation (Figure 3, Figure 4, Figure 5 and Figure 6). Soybean plants cultivated under deficient WR exhibited lower macronutrient accumulation values than those cultivated under full WR. Additionally, N supplementation promoted significant alterations in total biomass accumulation and the contents of N, magnesium (Mg), and sulfur (S). Although phosphorus (P), potassium (K) and calcium (Ca) were not significantly affected by N application alone, there was an interaction among the other factors (Table 1).

Variations in N accumulation were observed from the R1 stage onwards. Soybean plants under full WR and with N fertilization showed a maximum N accumulation at 85 DAE, when soybean plants were at the R8 stage (Figure 3A). Plants without N fertilization and under full WR exhibited maximum N accumulation at 82 DAE (R8) (Figure 4A). The greatest N accumulation occurred at 85 DAE (R8) in plants under deficient WR treatment and without N application and at 70 DAE (R7) with N supplementation (Figure 5A and Figure 6A).

Soybean seed yields were significantly altered by water regime differentiation. Under deficient WR, plants showed N accumulation of 170 kg ha^−^^1^ and 145 kg ha^−^^1^, with HIs of 0.73 and 0.86 with and without N fertilization, respectively. Under full irrigation, the values were found to be 383 kg ha^−^^1^ with N supplementation, having an HI of 0.89, and 309 kg ha^−^^1^ without N supplementation, with an HI of 0.87, but there was no significant difference for N supplementation (Appendix A).

The total P accumulation did not differ significantly by N fertilization but was significantly affected by variations in WR. Soybean plants showed a maximum P accumulation of 19 kg ha^−^^1^ under full WR, with and without N fertilization (Figure 3B and Figure 4B), while values of 12 kg ha^−^^1^ and 9 kg ha^−^^1^ were determined for plants under deficient WR, with and without N fertilization, respectively (Figure 5B and Figure 6B). Additionally, the HIs for P content in seeds were 0.93 and 0.80 for plants under full WR with and without N fertilization, respectively. Plants cultivated with deficient WR with and without N fertilization exhibited the same HI of 0.83 (Appendix A).

Plants under deficient WR and supplemented with N showed higher K values than unsupplemented plants. Ca accumulation was regulated by WR, showing a significant decrease in Ca content in plants under deficient WR conditions and without N supplementation. The maximum estimated Mg accumulations were 58 kg ha^−^^1^ and 40 kg ha^−^^1^ in plants with and without N fertilization under full WR, and 41 kg ha^−^^1^ and 22 kg ha^−^^1^ with and without N fertilization in deficient WR, respectively. The total Mg accumulation was considerably higher under full WR and when supplemented with N in both WRs, with significant increases in Mg accumulation (Appendix A).

The highest values of S accumulation were recorded in N-fertilized soybean plants under full WR. In plants under full WR treatments, the absence of N supplementation reduced the maximum S accumulation by 49% compared to that of N-fertilized plants (Figure 3F). Under deficient WR, soybean plants without N supplementation showed a 70% reduction in S accumulation compared to plants with N fertilization. The HI for S accumulation varied significantly in response to WR and N fertilization. The HIs for S in plants under full WR were 0.50 and 0.75 with and without N fertilization, respectively. In contrast, the HIs for S in plants under deficient WR were 0.66 and 0.50 with and without N supplementation, respectively.

The accumulation of macronutrients in soybean plants followed the order N > K > Ca > Mg > P > S, regardless of the treatment. The macronutrient contents in soybean plants under full WR and with N fertilization were N (349 kg ha^−^^1^) > K (127 kg ha^−^^1^) > Ca (107 kg ha^−^^1^) > Mg (74 kg ha^−^^1^) > P (23 kg ha^−^^1^) > S (16 kg ha^−^^1^). When full WR was not combined with N fertilization, the following values of macronutrient accumulation in soybean plants were recorded: N (299 kg ha^−^^1^) > K (129 kg ha^−^^1^) > Ca (87 kg ha^−^^1^) > Mg (57 kg ha^−^^1^) > P (24 kg ha^−^^1^) > S (8 kg ha^−^^1^). In soybean plants under deficient WR with N supplementation, the accumulation of macronutrients was N (217 kg ha^−^^1^) > K (85 kg ha^−^^1^) > Ca (65 kg ha^−^^1^) > Mg (37 kg ha^−^^1^) > P (13 kg ha^−^^1^) > S (7 kg ha^−^^1^). When N fertilizer was not used, soybean plants under deficient WR showed the following result: N (108 kg ha^−^^1^) > K (62 kg ha^−^^1^) > Ca (30 kg ha^−^^1^) > Mg (24 kg ha^−^^1^) > P (9 kg ha^−^^1^) > S (4 kg ha^−^^1^).

### 2.4. Micronutrient Accumulation

In the current study, water regime (WR) and N fertilization had a significant effect on the accumulation of micronutrients (Cu, Fe, Mn, Zn, and B) (Table 2 and Figure 7, Figure 8, Figure 9 and Figure 10) throughout the soybean cycle. Cu was the least accumulated nutrient by soybeans throughout the cycle for all treatments, while Fe was the most accumulated micronutrient. Overall, soybean plants under deficient WR showed lower values of Cu accumulation, especially when plants received N fertilization. The maximum accumulation of Cu occurred at R7 (70 and 76 DAE) under full WR, with and without N fertilization (77 and 79 g of Cu ha^−^^1^, respectively) (Figure 7A and Figure 8A). Conversely, the maximum accumulation of Cu in the deficient WR (30 g ha^−^^1^ and 45 g ha^−^^1^_,_ with and without N fertilization, respectively) occurred between R5 and R7 (60 and 71 DAE). The HIs for Cu under full WR were 0.73 and 0.75 with and without N fertilization, respectively. In the deficient WRs, the HIs for Cu were 0.59 and 0.60 with and without N supplementation, respectively.

The maximum Fe accumulation in soybean plants under full WR occurred at the R7 stage (77 DAE), showing values of 562 g ha^−^^1^ (HI = 0.66) and 535 g ha^−^^1^ (HI = 0.43) for Fe accumulation with and without N supplementation, respectively (Figure 7B and Figure 8B). In soybean plants under deficient WR, the maximum accumulation of Fe occurred at stages R8 (79 DAE) and R5 (58 DAE) with and without N fertilization, respectively. Fe accumulations in soybean plants under deficient WR with and without N supplementation were 383 g ha^−^^1^ (HI = 0.26) and 213 g ha^−^^1^ (HI = 0.54), respectively (Appendix A).

Plants under deficient WR without N fertilization showed lower Mn accumulation than the plants from other treatments (Appendix A). Mn accumulations in soybean plants under full WR were 181 g ha^−^^1^ (HI = 0.59) and 126 g ha^−^^1^ (HI = 0.48) with and without N fertilization, respectively. In contrast, Mn accumulations in plants under deficient WR were 122 g ha^−^^1^ (HI = 0.21) and 52 g ha^−^^1^ (HI = 0.50) with and without N supplementation, respectively.

The lowest values of Zn accumulation were recorded only in soybean plants under deficient WR without N fertilization. Under full WR, the maximum Zn accumulation was found to be 245 g ha^−^^1^ (stage R7; 60 DAE) and 216 g ha^−^^1^ (stage R8; 84 DAE) in plants under full WR with and without N supplementation, respectively, while under deficient WR, maximum Zn accumulation values of 131 g ha^−^^1^ (stage R7; 75 DAE) and 85 g ha^−^^1^ (stage R5; 57 DAE) were recorded in plants with and without N fertilization, respectively (Figure 9D and Figure 10D).

The water regime influenced the accumulation of B in soybean plants, with the lowest results observed in plants under deficient WR, which showed reductions of 33% and 36% with and without N fertilization, respectively, compared to plants under full WR in the same conditions. Under full WR, B accumulation in soybean plants under full WR with and without N supplementation was 154 g ha^−^^1^ (HI = 0.62) and 145 g ha^−^^1^ (HI = 0.60), respectively, while under deficient WR, B accumulation with and without N fertilization was 69 g ha^−^^1^ (HI = 0.36) and 62 g ha^−^^1^ (HI = 0.42), respectively (Appendix A).

The micronutrient accumulation in soybean plants under full WR and N fertilization followed the decreasing order: Fe (1.292 kg ha^−1^) > Mn (0.431 kg ha^−1^) > B (0.216 kg ha^−1^) > Zn (0.189 kg ha^−1^) > Cu (0.072 kg ha^−1^). In plants under full WR and without N supplementation, the following order of micronutrient accumulation was recorded: Fe (0.864 kg ha^−1^) > Mn (0.232 kg ha^−1^) > Zn (0.206 kg ha^−1^) > B (0.182 kg ha^−1^) > Cu (0.074 kg ha^−1^). In plants under deficient WR and N supplementation, the order of micronutrient accumulation was Fe (0.773 kg ha^−1^) > Mn (0.299 kg ha^−1^) > Zn (0.172 kg ha^−1^) > B (0.143 kg ha^−1^) > Cu (0.052 g ha^−1^). For plants under deficient WR without N fertilization, the following decreasing order of micronutrient accumulation was recorded: Fe (0.249 kg ha^−1^) > B (0.116 kg ha^−1^) > Mn (0.109 kg ha^−1^) > Zn (0.051 kg ha^−1^) > Cu (0.036 kg ha^−1^).

### 2.5. Seed Yield, Protein and Oil Percentages

The seed yield (seed dry biomass) of plants under full WR (5730 kg ha^−1^) was higher than that of plants under deficient WR (2231 kg ha^−1^) (Table 1); however, there was no significant difference when considering N supplementation (Appendix A). The seed yield and harvest index (HI) were 6070 kg ha^−1^ and 0.55, respectively, with N fertilization in plants under full WR. Without N supplementation, the seed yield was 5392 kg ha^−1^, and the HI was 0.51. These values are above the average yield of 3087 kg ha^−1^ recorded for the state of Piauí, Brazil, in the 2019/2020 season [19]. The seed yield was around 2110 kg ha^−1^ (HI of 0.38) and 2352 kg ha^−1^ (HI of 0.38) with and without N supplementation, respectively, in plants grown under deficient WR.

The results for protein content in seeds at the R8 stage showed no significant differences across treatments. In contrast, the oil content in plants supplemented with N under full WR (21.9%) was higher than that of plants under deficit irrigation (17.5%), and plants under deficit irrigation and N fertilization (17.5%) showed lower oil content in relation to plants without added N (20.4%) (Figure 11).

## 3. Discussion

Numerous reports have shown that flowering and seed filling are the phases that are most sensitive to water deficits during the soybean cycle [10,11,20]. Research conducted in Argentina and the USA, combining different sowing times, cultivars, water regimes and N availability, defined the critical period for seed number determination as R3–R6 [21]. In the present study, the area subjected to deficient WR (50% ETc) showed soil moisture values below the critical moisture content, indicating that the plants were subjected to water deficit conditions (Figure 1). Additionally, even with soil moisture above the permanent wilting point (PWP), soybean plants were not in the full physiological conditions to express their maximum seed productivity capacity [22].

Nutrient accumulation and partitioning in modern soybean cultivars have an estimated maximum biomass accumulation ranging between 9201 and 10,586 kg ha^−1^ at the R7 stage [10]. In experiments with high-yielding soybean cultivars and an ample supply of fertilizer N, the estimated average for accumulated dry matter (ADM) was 13,200 kg ha^−1^; however, when the crop relied on biological N_2_ fixation and indigenous soil N, the estimated ADM was 11,600 kg ha^−1^ [23]. Estimated accumulations under deficient WR conditions were lower than these values and occurred close to R5 when supplemented with N, and at R1 when not supplemented with N. Under full irrigation and without N fertilization, the results were similar to those reported values, demonstrating that plant development is directly linked to adequate water availability.

Biomass accumulation has been cited as an important trait in plants growing under water limiting conditions [24,25]. Herein, approximately 55% of the total DM accumulation occurred up to R5 for treatments under a full water regime and N supplementation (Figure 2A), while 76% accumulated at the same stage when no N was supplemented (Figure 2B). Plants under deficient WR showed 49% and 56% of the total accumulation (with and without N, respectively) (Figure 2C,D) at R4. As a result, plants under full WR displayed seed yields higher than those subjected to deficient WR, without significant differences regarding N supplementation. Our results agree with previous studies with soybean plants that reported 50% DM production after R4, concluding that higher seed yields in soybean may be associated with greater accumulation after R5 rather than early accumulation [24,25].

N supplementation provided an increase in yield, but this increase was not significant. Plants without N supplementation had BNF as the main source of N, but indigenous N in soil might have contributed. Aiming to evaluate the degree to which indigenous soil N supplements modulates the yield difference between N-fertilized plants, researchers have estimated the N supply from indigenous N in soil under different environments ranging from 63 to 208 kg N ha^−1^ [23]. An important point to emphasize is that in productive soybean systems, this higher indigenous N supply combined with BNF cannot meet the N requirement of the crop [17], as BNF decreases as the contribution from soil N supply increases [26].

In addition, our results are in discordance with previous studies that reported the maximum N accumulation in indeterminate growth soybean as typically occurring at R7 [10,27]. Herein, soybean plants with and without N fertilization and under full WR showed maximum estimated accumulation at the R8 stage. Under water deficit conditions without N application, the maximum accumulation also occurred at R8. However, with N supplementation occurring at R6 (70 DAE) (Appendix A), these results indicate an advance of maximum N accumulation in soybean plants supplemented with N under water deficit. Additionally, plants are not expected to obtain BNF contributions due to high-dose N supplementation because N fertilization may have negative effects on bacterial adhesion, nodule development, and nitrogenase activity. Similar results showing earlier N accumulation have been reported for high-yielding soybean cultivars (>5 Mg ha^−1^), where there was an asynchrony; N accumulation peaked earlier in the full-N treatment than in the zero-N treatment [17].

The present study recorded harvest indexes (HI) between 61% and 75% for P in plants under full or deficient WR. These values are lower than those recommended for medium- and high-income areas, where HI values are approximately 80% [11]. It is worth mentioning that the soil of the study area is tropical, with 1:1 clay minerals, and contains Fe and Al oxides that contribute to high P fixation [28]. In addition to P, the accumulation of K was significantly reduced in soybean plants under deficient WR compared to that recorded for full WR, regardless of N supplementation. K is a nutrient directly involved in the osmotic control of cells and the opening and closing of stomata [3]. The transport of K from soil to plant roots is largely performed by diffusion, a process affected by water availability. Therefore, the low water supply can reduce the nutrient content absorbed by the root system. The water deficit imposed on plants under deficient WR may have reduced K uptake by the roots and, consequently, K presence in other plant tissues. Considering the importance of K, the supplementation of this nutrient in soybean plants improves physiological responses under water deficit conditions, allowing for photosynthetic recovery after the rehydration period [29].

Compared with other nutrients, there was a low redistribution of Ca in the seeds at the R8 stage. This behavior is related to the structural function of Ca and its low mobility in plants [30]. On the other hand, the values of Mg accumulation in seeds were higher than those reported in the literature, especially under full WR [10,11,31]. The higher Mg accumulation may have been the result of efficient absorption through foliar fertilization, which also promoted benefits for gas exchange and antioxidant metabolism [32]. However, S was less required by soybean plants than other macronutrients, with HI values of approximately 60%, a result that corroborated previous reports from [11]. S and N may have a synergistic relationship, classified in the same biochemical function group [30], which explains the lower S values in soybean plants not supplemented with N.

The soybean plants displayed significant alterations in the accumulation of micronutrients (Cu, Fe, Mn, Zn, and B) due to water regimes and N fertilization. In general, Cu accumulated less throughout the soybean cycle in all treatments. Higher HI values, except for Mg, S and Cu, were recorded in plants that received N fertilization under full WR. These results indicate that N fertilization leads to a greater accumulation of nutrients in storage organs compared to that observed in plants not supplemented with N. However, there was no close relationship with greater translocation of nutrients to the seeds, suggesting that soybean quality is not only exclusive to nutrient uptake but also depends on environmental conditions.

The highest losses (50% reductions in nutrient accumulation) in the total accumulation were recorded for Mg and S in plants under deficient WR and N supplementation (Figure 5) compared to plants under full WR (Figure 3 and Figure 4). Furthermore, 75% and 64% decreases in the accumulation of Zn and N were observed in plants under full WR without N fertilization. In comparison, the losses were 9% for Zn and 38% for N in plants under deficient WR and N fertilization. However, plants treated with N fertilization exhibited greater amounts of nutrients in the tissues (leaf, stem, and reproductive structures) compared to those that did not receive N, especially under deficient WR. These findings indicate nutrient partitioning in plants exposed to water deficit under N fertilization, a possible advantage of increasing N assimilation to mitigate the harmful effects of water deficit.

The seed yield of soybean plants under full WR was higher than that recorded in plants under deficient WR, without any alteration by N fertilization (Appendix A). These data reinforce the critical role of water in growth and yield parameters, particularly the redistribution of DM from vegetative to reproductive organs. Additionally, the data suggest that low seed yield in plants under deficient WR mostly results from a combination of water deficit and high temperatures [24]. A maximum temperature of 36 °C was recorded during the experiment, a condition superior to the ideal maximum for soybean growth and development [33].

The seed protein content was not altered by the treatments, but the oil content under full WR was increased by N fertilization, suggesting that N-supplemented plants activated genes for transcription factors (TFs) and enzymes for the biosynthesis of fatty acids and triacylglycerol [34]. Similar findings were reported in research with N supplementation in soybean plants, where an increase in oil content from seeds of N-fertilized plants occurred [5].

Taken together, the findings invalidate our first hypothesis, highlighting that N application does not result in gains of soybean plants under drought stress. The data clearly validate the second hypothesis, demonstrating that the combination of N fertilization and water regime modifies the uptake and partition of nutrients by soybean plants in the Cerrado of northeastern Brazil.

## 4. Materials and Methods

### 4.1. Growth Conditions and Treatments

The experiment was carried out in the experimental field of Embrapa Mid-North in Teresina, Piauí, Brazil (05°05′ S, 42°49′ W), from July to October 2019. According to the Köppen classification, the region has a tropical climate with a dry winter season (Aw’ type). During the experiment, the average, maximum, and minimum air temperatures and accumulated rainfall were 29 °C, 21 °C, 36 °C, and 8 mm, respectively [35] (Figure 12).

The chemical characterization of soil from the experimental area, classified as eutrophic Ultisol [36] of loam-sandy texture, is presented in Table 3. The soybean cultivar was BMX Bônus (8579 IPRO), which has an indeterminate growth habit and maturation group 7.9. Before sowing, the seeds were inoculated with *Bradyrhizobium japonicum* in a proportion of 100 mL of inoculant to 7.0 kg of seeds. Sowing was performed manually, distributing 20 seeds per meter in the planting furrow. Thinning was performed after germination to maintain 10 to 12 plants per meter. The plots were composed of 18 lines, 6.0 m long and spaced 0.5 m apart.

Fertilization was performed based on the chemical attributes of the soil. Micronutrients were applied through foliar fertilization as follows: manganese (Mn) (75 g ha^−1^ at V3, 150 g ha^−1^ at V7 and R1, and 75 g ha^−1^ in R3); zinc (Zn) (40 g ha^−1^ in V3 and 80 g ha^−1^ in V7 and R1); copper (Cu) (25 g ha^−1^ in V7 and R1); cobalt (Co) (4 g ha^−1^ in V3); molybdenum (Mo) (40 g ha^−1^ in V3); nickel (Ni) (40 g ha^−1^ in V3); and boron (B) (50 g ha^−1^ in V7, R1, and R3). Phosphorus (P) was applied only during planting using single superphosphate at a rate of 120 kg ha^−1^ of P_2_O_5_. KCl (potassium chloride) was used for potassium (K) supplementation, applied during planting (96 kg ha^−1^ of K_2_O) and at the V8 stage (96 kg ha^−1^ of K_2_O). Magnesium (Mg) was supplemented at V7 (300 g ha^−1^) and R3 (300 g ha^−1^) using foliar fertilization. The preventive management of pests and diseases was performed through chemical control and manual weeding.

The water regimes (WR) were applied using a conventional fixed sprinkler system, with sprinklers spaced at 12.0 × 12.0 m. Irrigation management was performed based on the replenishment of crop evapotranspiration (ETc) estimated through the coefficients of irrigation for soybeans proposed by FAO. Reference evapotranspiration (ETo) was calculated using the Penman–Monteith method [37] and daily climate data from the INMET meteorological station.

The plants were initially grown under a full water regime (100% ETc). From the reproductive phase R1 [38], at 35 days after emergence (DAE), water regimes were differentiated and applied based on crop evapotranspiration (ETc) (100 and 50% of evapotranspiration), i.e., the ideal situation of a full water regime and a water stress condition. The soil moisture was monitored using soil moisture probes, model CS616 (Campbell), with three rods positioned in each water regime, two at 0.0–0.3 m and one at 0.3–0.6 m depth. The moisture data were recorded continuously using a CR1000 datalogger. The experimental area was subdivided into three strips, as follows: strip 1, with soybean under irrigation deficit (50% ETc); strip 2, with soybean under full irrigation (100% ETc); and, between the two strips, a third that corresponds to the free area cultivated with soybeans as borders, avoiding contact with adjacent water regimes.

Two conditions were established for N fertilization: (1) plants without mineral N supplementation; and (2) plants with N fertilization at a rate of 1000 kg N ha^−1^ using urea (46% of N), which was split as follows: 20 kg at planting, 80 kg at V3, 100 kg when beginning V5, 200 kg at R1, 300 kg at R3, and the remaining 300 kg ha^−1^ at R5. A rate of 1000 kg N ha^−1^ was established to meet the N demand throughout the soybean cycle, with an expected yield of 6.0 Mg ha^−1^, considering that 80 kg ha^−1^ is necessary for each Mg of soybean produced [39], and the efficiency of N fertilization is approximately 43% [40].

### 4.2. Plant Harvests and Trials

The experimental design was randomized blocks in a split-split plot arrangement, with the plots corresponding to the water regimes (full and deficient), the split plots to N fertilization (with and without application of N fertilizer), and the split-split plots to the harvest times of soybean plants (16, 23, 30, 37, 44, 58, 65, 79 and 86 DAE), with three replicates.

The first harvest was carried out at the V3 stage by collecting five plants from each treatment to quantify dry biomass accumulation, nutritional analysis, and seed oil and protein content. The plants were partitioned into stems, leaves, reproductive structures, and seeds and washed in running water, followed by distilled water [41]. The dry biomass of each part was quantified after drying in a forced air oven at 65 °C until a constant mass was reached.

The dry biomass and seed yield were estimated, both per area, based on the stand containing ten plants per linear meter and expressed in kg ha^−1^. After drying, the samples were processed and used to analyze macronutrients (N, P, K, Ca, Mg, and S) and micronutrients (Cu, B, Fe, Mn, and Zn) according to the methodology described by Miyazawa et al. [42] after nitroperchloric digestion: P was determined by colorimetry; K by flame photometry; S by turbidimetry; Ca, Mg, Cu, Fe, Mn, and Zn by atomic absorption spectrophotometry; N by the Kjeldahl method; and B by calcination. Their accumulation was calculated by multiplying the nutrient content by the biomass of the organ. The harvest index (HI) was calculated from the accumulated value of the nutrient in relation to the content in seeds. Oil was extracted in petroleum ether according to the Goldfish method [43]. Protein (crude) was determined by the Kjeldahl method [43].

### 4.3. Statistical Analysis

The data obtained in each collection were tested by analysis of variance. The dry matter production and nutrient accumulation in each plant structure as a function of days after emergence were analyzed using nonlinear regression. The significance of the parameters and the coefficient of determination (R^2^) were observed to choose the regression models. The data were fitted to nonlinear regression models, Gaussian (a*exp[−0.5*(x − x0)/b]^2^) and Lorentizian (a/[1 + (x − x0)/b]^2^).

The value of the inflection point (IP) in the regressions where the Gaussian and Lorentizian models were used was obtained through the following formula: IP = X0 − b, where IP corresponds to the value of X in which the curvature of the adjusted model changes sign (that is, it corresponds to the value of X in DAE at which the daily accumulation rate, although positive, begins to decrease). The t test at 5% probability was used to evaluate the effect of treatments on dry matter, seed yield, and oil and protein content in the seeds.

## 5. Conclusions

Water stress severely limits soybean performance from R1 to R6, highlighting the low biomass accumulation, seed yield, and oil content. Our findings clearly demonstrate that the patterns of nutrient accumulation, partitioning, and redistribution are disturbed in drought-stressed plants. Nitrogen fertilization is able to advance and increase the maximum N accumulation but does not result in significant yield gains in the environment evaluated. This study provides important information for the cultivation of soybean crops in semiarid regions with nutritional limitations.

## Figures and Tables

**Figure 1 plants-12-01898-f001:**
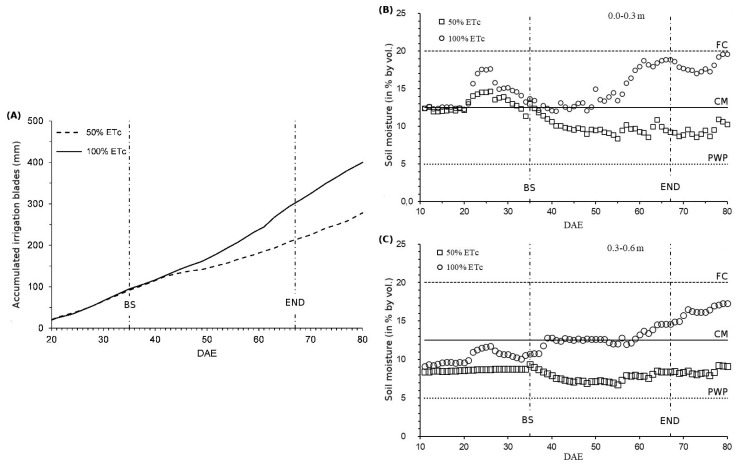
Accumulated irrigation depths (**A**) and soil moisture content measured during the soybean cycle in response to water regimes (50% and 100% of ETc) applied to 0.0–0.3 m (**B**) and 0.3–0.6 m (**C**) soil layers. DAE = days after emergence. FC = field capacity. CM = critical moisture. PWP = permanent wilting point. BS = beginning of stress (differentiating between WRs at 35 DAE). END = end of the application of the differentiated irrigation depths (67 DAE).

**Figure 2 plants-12-01898-f002:**
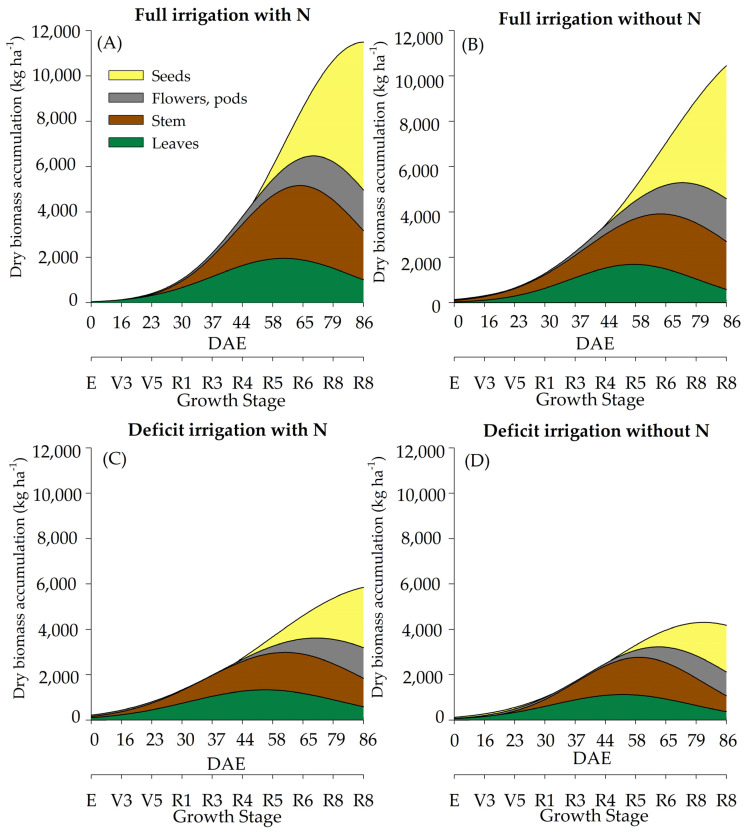
Total dry biomass of soybean plants under full irrigation with N (**A**), full irrigation without N (**B**), deficit irrigation with N (**C**), and deficit irrigation without N (**D**).

**Figure 3 plants-12-01898-f003:**
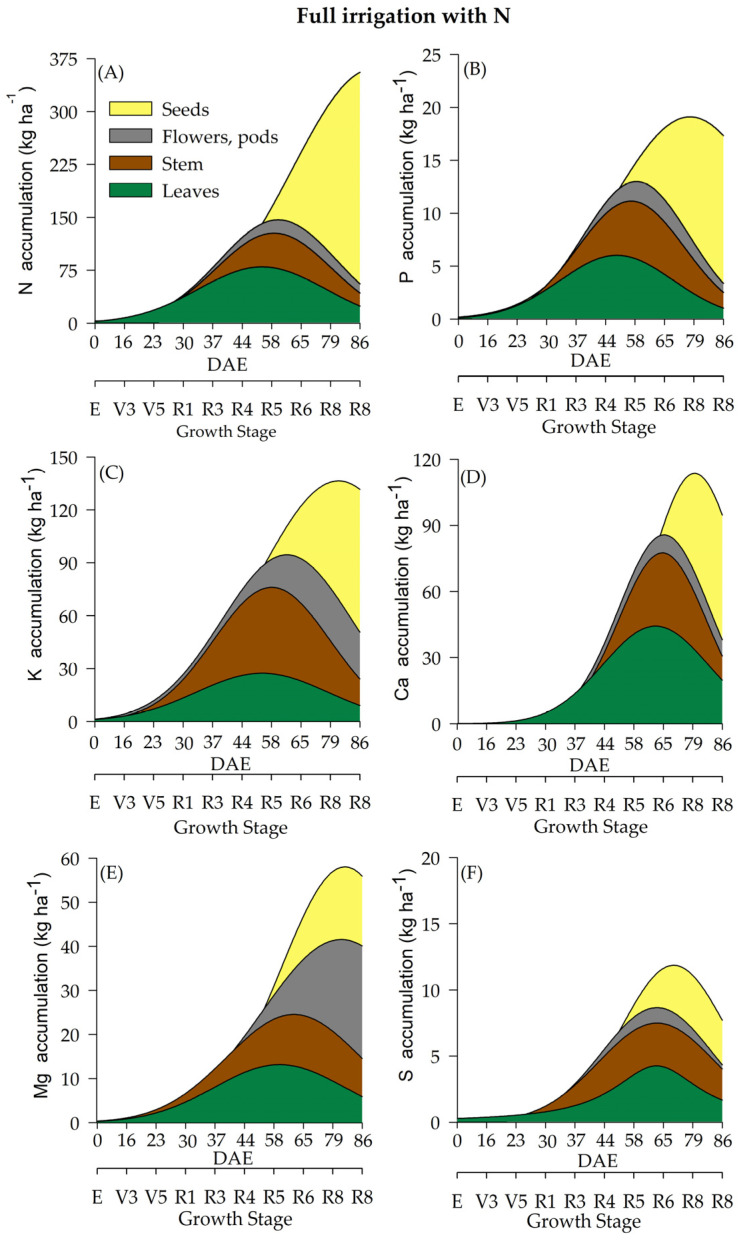
Total accumulation of macronutrients [nitrogen-N (**A**), phosphate-P (**B**), potassium-K (**C**), calcium-Ca (**D**), magnesium-Mg (**E**) and sulfur-S (**F**)] under a full water regime and N fertilization throughout the soybean cycle.

**Figure 4 plants-12-01898-f004:**
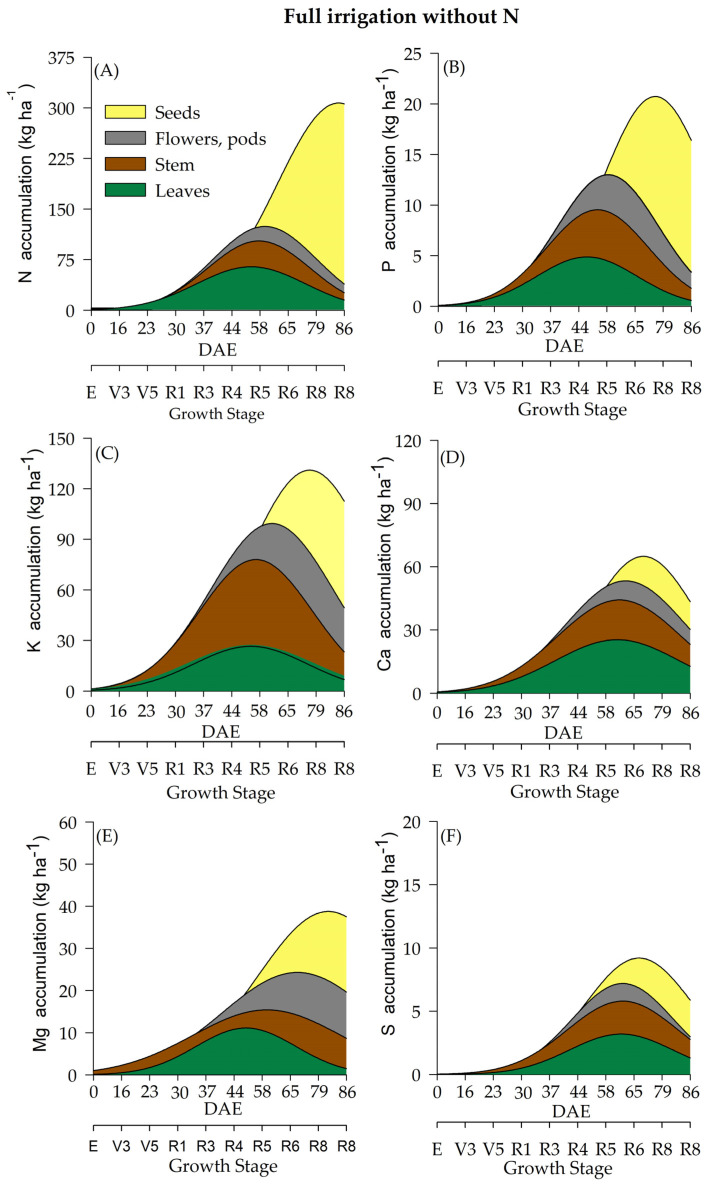
Total accumulation of macronutrients [nitrogen-N (**A**), phosphate-P (**B**), potassium-K (**C**), calcium-Ca (**D**), magnesium-Mg (**E**) and sulfur-S (**F**)] under a full water regime and without N fertilization throughout the soybean cycle.

**Figure 5 plants-12-01898-f005:**
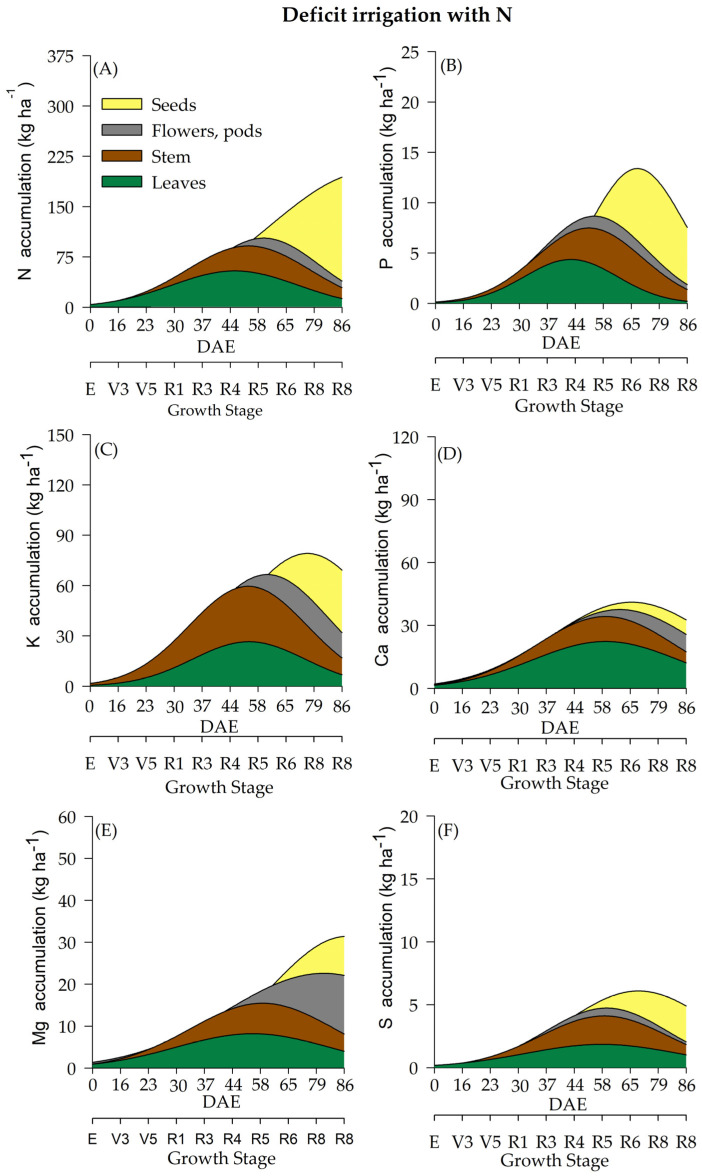
Total accumulation of macronutrients [nitrogen-N (**A**), phosphate-P (**B**), potassium-K (**C**), calcium-Ca (**D**), magnesium-Mg (**E**) and sulfur-S (**F**)] under a deficient water regime and with N fertilization throughout the soybean cycle.

**Figure 6 plants-12-01898-f006:**
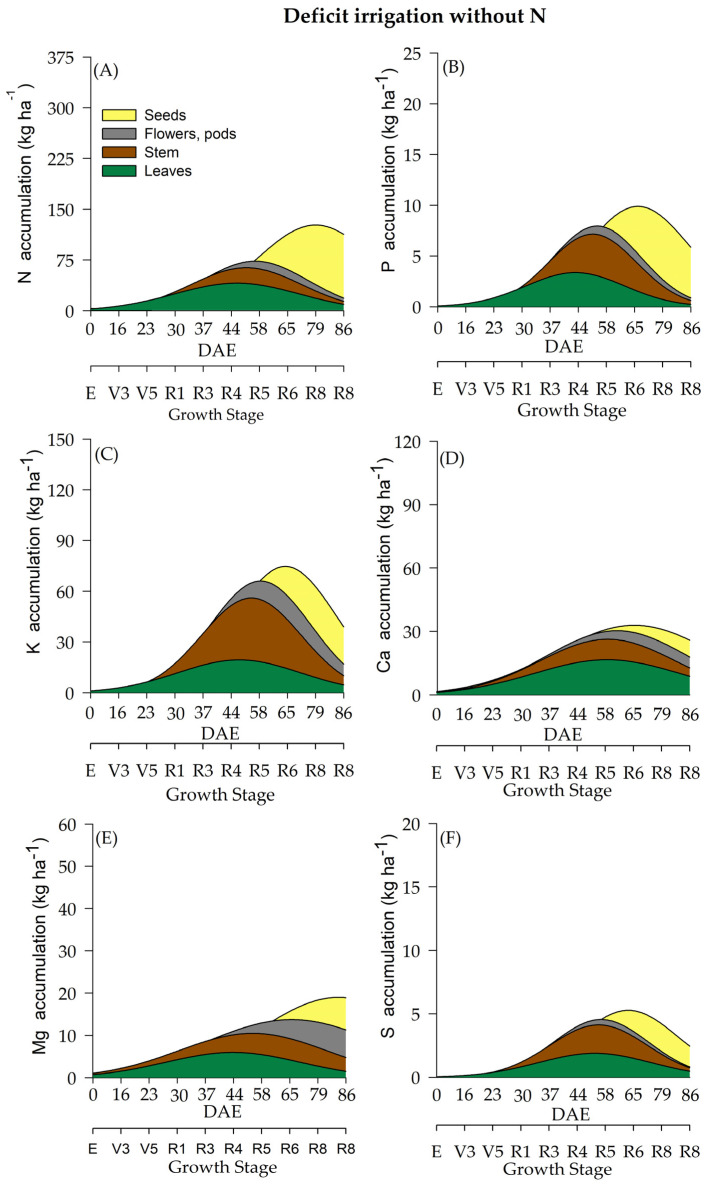
Total accumulation of macronutrients [nitrogen-N (**A**), phosphate-P (**B**), potassium-K (**C**), calcium-Ca (**D**), magnesium-Mg (**E**) and sulfur-S (**F**)] under a deficient water regime and without N fertilization throughout the soybean cycle.

**Figure 7 plants-12-01898-f007:**
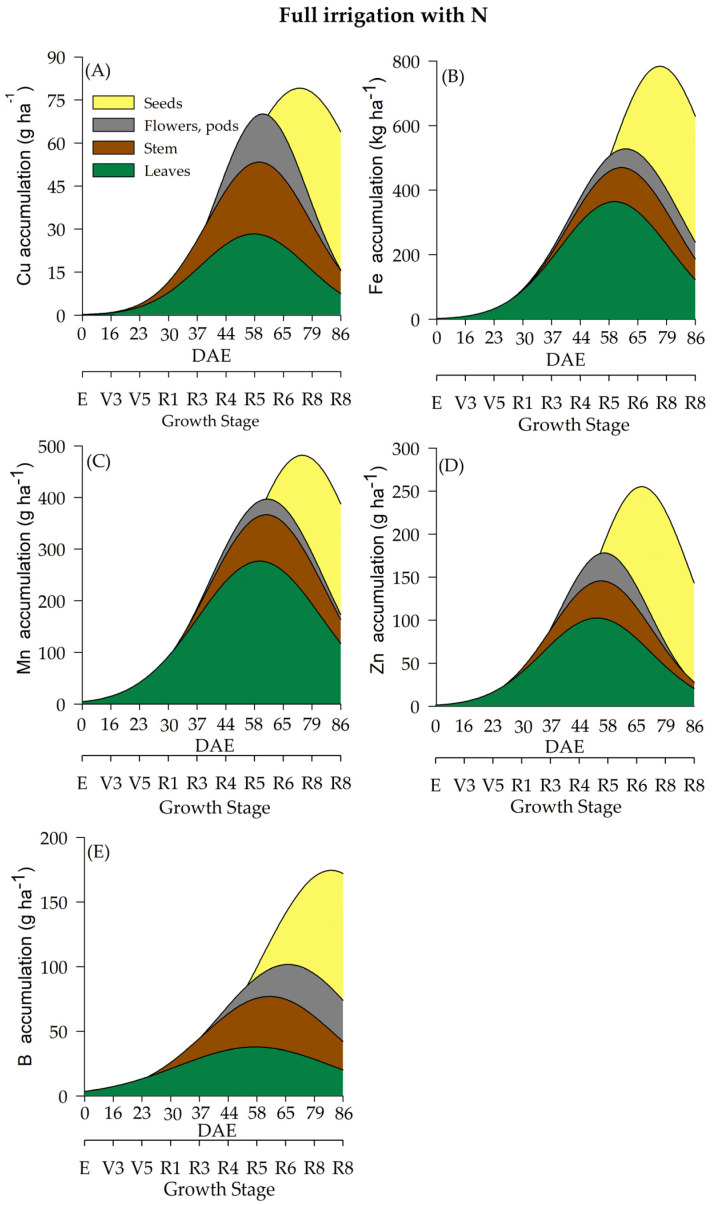
Total accumulation of micronutrients [copper-Cu (**A**), manganese-Mn (**B**), iron-Fe (**C**), zinc-Zn (**D**) and boron-B (**E**)] under a full water regime with N fertilization throughout the soybean cycle.

**Figure 8 plants-12-01898-f008:**
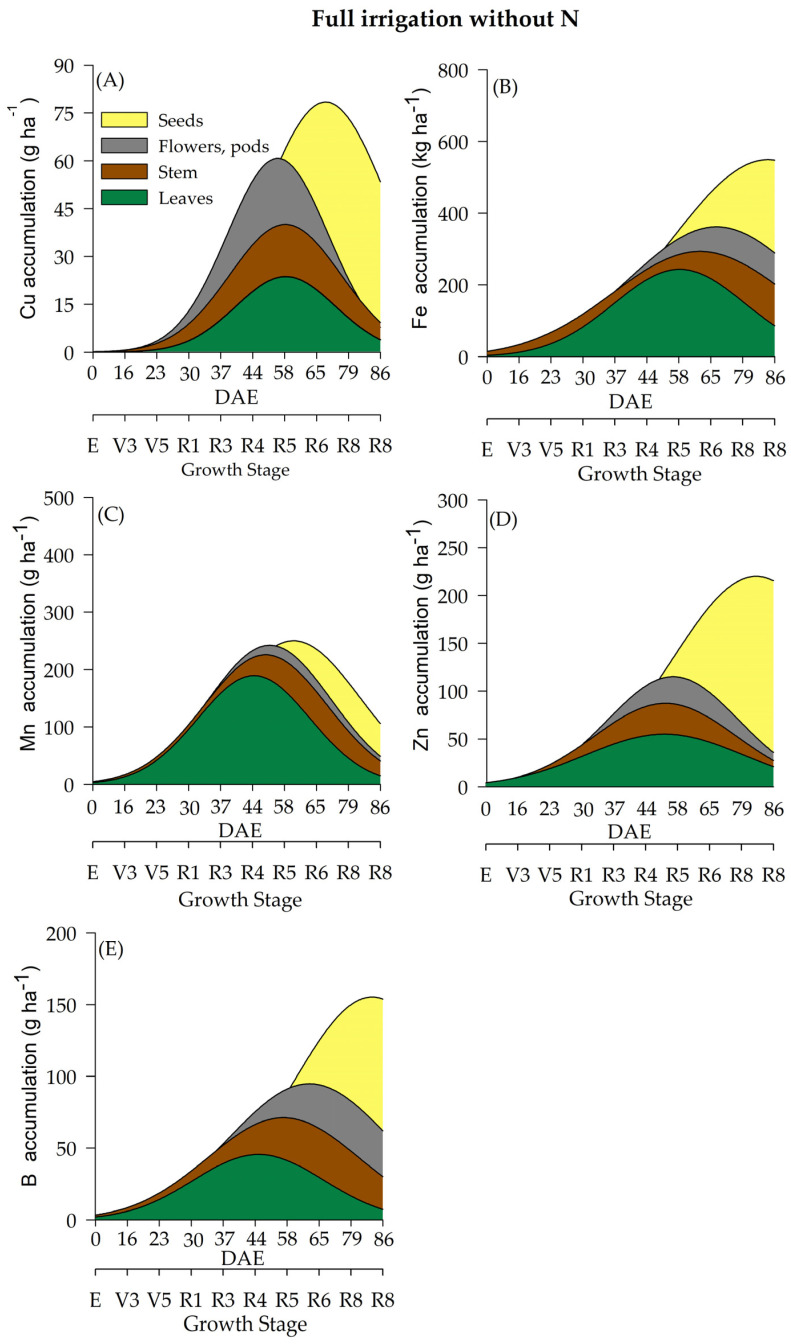
Total accumulation of micronutrients [copper-Cu (**A**), manganese-Mn (**B**), iron-Fe (**C**), zinc-Zn (**D**) and boron-B (**E**)] under a full water regime without N fertilization throughout the soybean cycle.

**Figure 9 plants-12-01898-f009:**
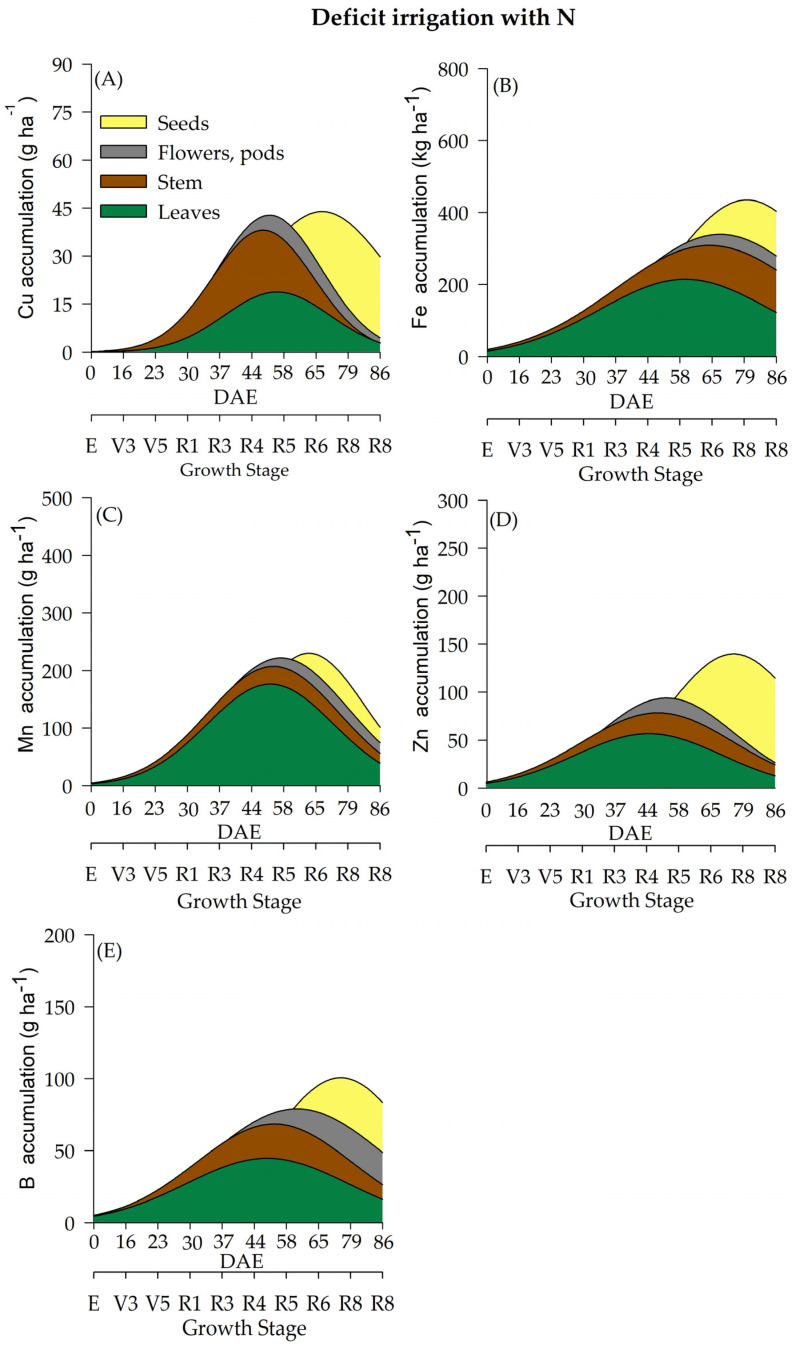
Total accumulation of micronutrients [copper-Cu (**A**), manganese-Mn (**B**), iron-Fe (**C**), zinc-Zn (**D**) and boron-B (**E**)] under a deficient water regime with N fertilization throughout the soybean cycle.

**Figure 10 plants-12-01898-f010:**
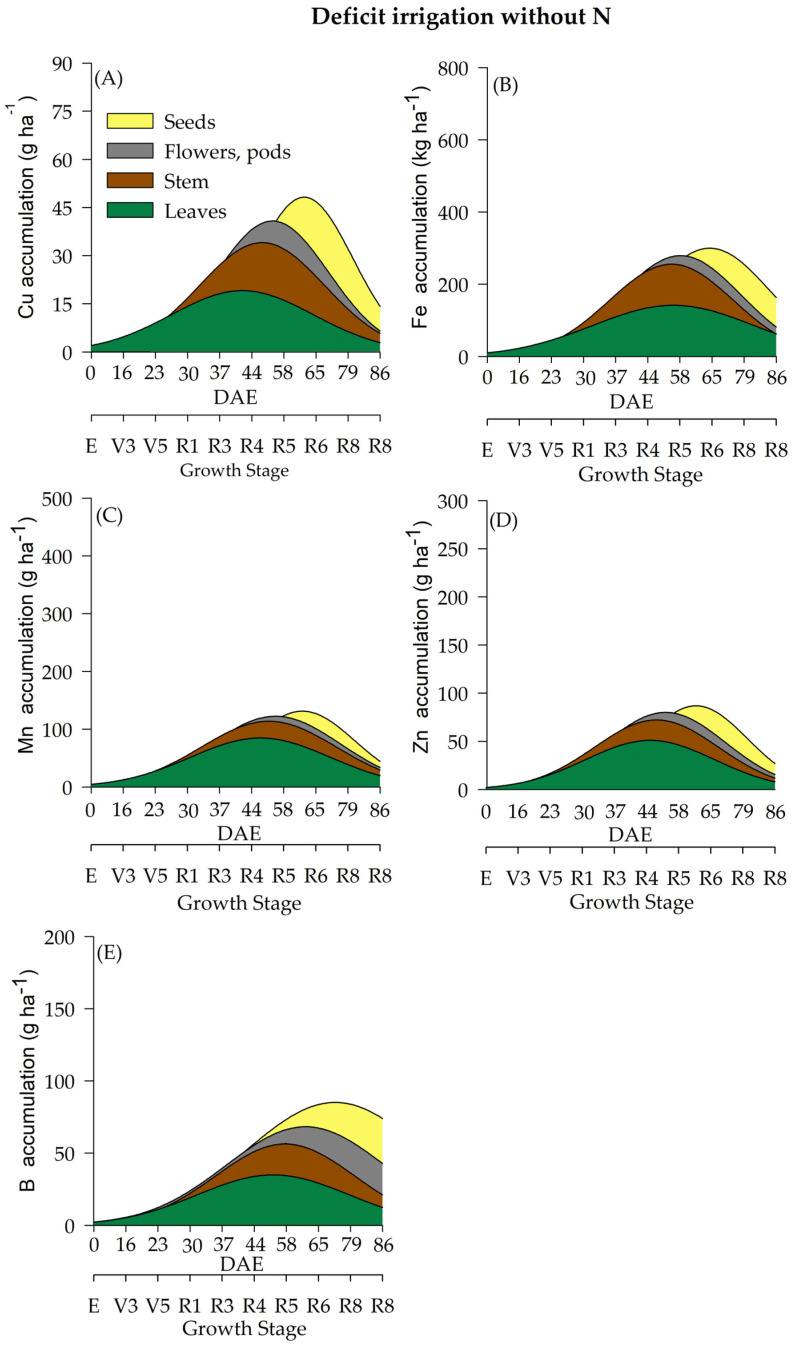
Total accumulation of micronutrients [copper-Cu (**A**), manganese-Mn (**B**), iron-Fe (**C**), zinc-Zn (**D**) and boron-B (**E**)] under a deficient water regime without N fertilization throughout the soybean cycle.

**Figure 11 plants-12-01898-f011:**
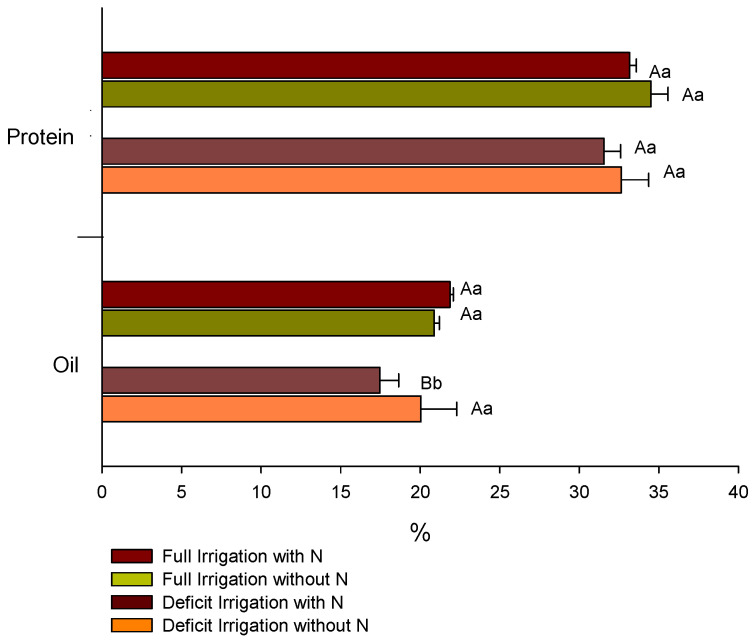
Protein and oil content in soybean seeds from different water regimes and nitrogen (N) treatments. Similar lowercase letters denote no significant difference concerning nitrogen fertilization (with N × without N), whereas similar uppercase letters denote significant differences.

**Figure 12 plants-12-01898-f012:**
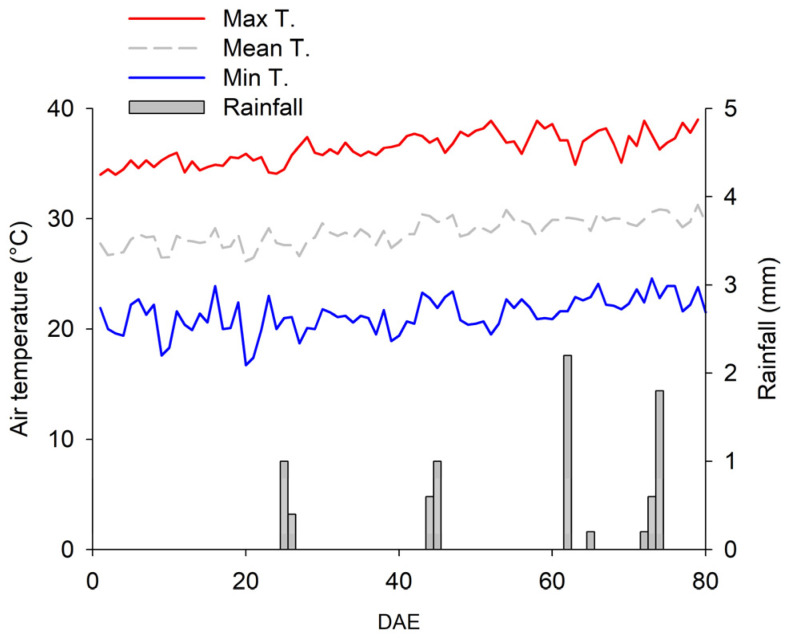
Maximum (Max T), minimum (Min T), and average air temperatures (Mean T) and rainfall during the experiment. DAE—days after emergence.

**Table 1 plants-12-01898-t001:** Analysis of variance of the effects of water regimes, with and without nitrogen supplementation, on soybean plants using mean values measured at R8 (physiological maturity) for dry biomass, macronutrient accumulation, seed yield, seed protein, and seed oil.

Source of Variation		ADM	N	P	K	Ca	Mg	S		Seed Yield	Seed Protein	Seed Oil
Design	df	F	F	F	F	F	F	F	df	F	F	F
Blocks	2	1.79	2.66	0.64	0.35	3.31 *	0.20	1.47	2	0.88	0.24	1.11
Treatment												
WR	1	1028.53 **	7908.64 **	8247.47 **	479.93 *	414.15 *	1014.90 **	64.62 *	1	220.03 **	1.63	8.45
Error1	2								2			
N	1	44.07 *	36.47 *	0.91	2.69	16.07	158.89 *	22.99 *	1	0.69	3.41	2.72
Error2	2								4			
DAE	8	485.87 **	392.25 **	401.13 **	323.88 **	195.78 **	408.48 **	143.66 **	-			
WR × N	1	0.05	1.26	17.24 **	10.39 *	0.97	0.24	11.60 **	1	3.07	0.01	13.66 **
WR × DAE	8	71.24 **	74.81 **	73.94 **	46.16 **	46.57 **	71.08 **	19.98 **	-			
N × DAE	8	7.76 **	3.71 **	1.05 **	2.44 *	5.44 **	17.41 **	7.23 **	-			
WR × N × DAE	8	2.04 *	3.87 **	10.51 **	3.12 *	3.31 *	7.23 **	3.69 **	-			
Error3	66											
Error		MS	MS	MS	MS	MS	MS	MS		MS	MS	MS
Blocks × Treatment	107	216,303.78	260.63	0.99	60.69	30.66	6.58	0.69	11	206,608.00	6.12	4.68
CV 1 (%) ^1^ =		8.55	3.34	3.30	11.47	13.94	8.71	30.25		10.27	4.97	7.80
CV 2 (%) ^2^ =		10.22	17.36	17.40	12.07	25.20	11.72	22.78		11.42	4.93	4.16
CV 3 (%) ^3^ =		12.84	14.64	12.44	13.14	17.33	14.04	20.96		-	-	-

df = degrees of freedom; F = F-statistic; MS = mean square, ADM = final aboveground biomass. Statistical significance is indicated by * = *p* value < 0.05; ** = *p* value < 0.01. ^1^ Coefficient of variation of the main plot. ^2^ Coefficient of variation of the split plots. ^3^ Coefficient of variation of the split-split plots.

**Table 2 plants-12-01898-t002:** Analysis of variance of the effects of water regimes with and without nitrogen supplementation on soybean plants using mean values measured at R8 (physiological maturity) for micronutrient accumulation.

Source of Variation		Cu	Fe	Mn	Zn	B
Design	df	F	F	F	F	F
Blocks	2	0.67	1.63 **	0.59	1.70	2.41
Treatment						
WR	1	854.53 **	5337.30 ***	207.09 *	6428.20 **	844.62 **
Error1	2					
N	1	39.39 *	216.97 *	594.98 ***	87.93 *	9.39 *
Error2	2					
DAE	7	484.09 ***	504.01 ***	231.31 ***	274.35 ***	657.51 ***
WR × N	1	0.02	0.09	1.18	28.46 ***	24.49 ***
WR × DAE	7	78.67 ***	99.79 ***	44.31 ***	80.08 ***	88.41 ***
N × DAE	7	4.69 ***	15.47 ***	22.95 ***	10.40 ***	3.08 *
WR × N × DAE	7	17.65 ***	13.54 ***	17.74 ***	41.41 **	10.26 ***
Error3	58					
Error		MS	MS	MS	MS	MS
Blocks × Treatment	95	17.75	934.52	388.64	164.77	18.80
CV 1 (%) ^1^ =		9.74	3.44	22.09	4.31	7.20
CV 2 (%) ^2^ =		7.98	8.08	8.98	10.24	10.05
CV 3 (%) ^3^ =		12.42	11.00	14.62	13.72	10.31

df = degrees of freedom; F = F-statistic; MS = mean square. Statistical significance is indicated by * = *p* value < 0.05; ** = *p* value < 0.01; *** = *p* value < 0.001. ^1^ Coefficient of variation of the main plot. ^2^ Coefficient of variation of split plots. ^3^ Coefficient of variation of split-split plots.

**Table 3 plants-12-01898-t003:** Chemical characterization and particle size of the soil at the 0–0.2 m and 0.2–0.4 m layers before implementing the experiment.

Depth	pH	OM	N	P	K^+^	Na^+^	Ca^2+^	Mg^2+^	Al^3+^
(m)	(CaCl_2_)	|-----g kg^−1^ -----|		mg dm^−3^	|------------------ cmol_c_ dm^−3^ -------------------|				
0.0 a 0.2	5.4	10.5	1.1	9.0	0.2	0.3	2.4	0.8	0.0
0.2 a 0.4	5.3	9.2	0.9	4.7	0.2	0.1	2.7	0.8	0.0
**Depth**	**H + Al**	**SB**	**CEC**	**BS**	**Clay**	**Silt**	**Sand**		
(m)	|------- cmol_c_ dm^−3^ ------|			|-%-| |------ (g kg^−1^) -----|					
0.0 a 0.2	1.3	3.7	5.0	74	150	110	730		
0.2 a 0.4	1.4	3.9	5.3	73	240	100	660		

OM: organic matter; N: nitrogen; P: phosphorus; K^+^: potassium; Na^+^: sodium; Ca^2+^: calcium; Mg^2+^: magnesium; Al^3+^: aluminum; H + Al: potential acidity; SB: sum of bases; CEC: cation exchange capacity; BS: base saturation.

## Data Availability

Not applicable.

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
