# Peer review of "Macro and Micro-Nutrient Accumulation and Partitioning in Soybean Affected by Water and Nitrogen Supply"

_plants, 2023, doi:10.3390/plants12091898_

Round 1
Reviewer 1 Report
It is a very interesting study with potentially applicable results to soybean production. The experiment is well conducted and analysed, the results are sufficient and are generally well presented. The discussion is mostly appropriate. I included several suggestions within the pdf, the manuscript can be approved with minor changes.

Author Response
We thank the excellent comments of the Reviewer which greatly contributed to improve the quality of manuscript. The larje marjority of them were completely accepted. We have now completed all of the changes requested and the responses to Reviewer is provided in PDF file. In the text, the revised portions of the manuscript are highlight with red font.

Reviewer 2 Report
The effect of drought stress on the formation of the mineral composition of plants has not been sufficiently studied. Soybean is a strategic crop. Therefore, the topic of this article is certainly relevant.
In the article, the authors cite 39 literary sources, more than 80% of which were published in the last 6 years. All sources refer to the soybean crop.
The introduction fully reflects the problem under study.
The purpose and the experimental part of the research are clearly stated.
The purpose and experimental part of the research is clearly stated.
In the section of materials and methods, the conditions of the experiment of growing soybeans and research methods are described in detail.
The experimental material is perfectly illustrated with 12 figures and 3 tables.
In the discussion the authors used more than 20 literature sources, which allowed them to substantiate the hypotheses put forward.
This work is of undoubted fundamental and practical importance.
The authors are advised to clarify whether the reference to work 3 (lines 38-40) in the sentence ".... a nutrient that plays a central role in 38 plant metabolism as a structural compound of amino acids, proteins, and chlorophylls, 39 thus playing a critical role in photosynthesis [3].
The statistical analysis is compelling.
After minor revision, the article may be recommended for publication in the journal Plants.
Author Response
We thanks the Reviewer comment. The unique sentence pointed was completely revised. The text was altered:
"Soybean is a crop with high demand for nitrogen (N), a nutrient that plays a central role as a structural component of amino acids, proteins, and chlorophylls. It also composes other N-containing compounds such as hormones and nucleic acids, thereby playing a critical role in cell function and plant metabolism."
Reviewer 3 Report
Line 18-19: Please use qualitative and quantitative word rather than variables.
Over all abstract is poorly written and no has important information presented in abstract. Therefore please re-write it.
Introduction is well written just try to avoid short para.
Please cite the below related articles.
Akhtar, K.; Wang, W.; Ren, G.; Khan, A.; Feng, Y.; Yang, G.; Wang, H., Integrated use of straw mulch with nitrogen fertilizer improves soil functionality and soybean production. Environment International 2019, 132, 105092 Akhtar, K.; Wang, W.; Khan, A.; Ren, G.; Afridi, M. Z.; Feng, Y.; Yang, G., Wheat straw mulching offset soil moisture deficient for improving physiological and growth performance of summer-sown soybean. Agricultural Water Management 2019, 211, 16-25. Results and discussion are well written but i suggest to add sub-heading in the discussion section same as results. Conclusion is very short, explain and make it attractive. Statistical analysis portion is missing add it.Author Response
We appreciate the comments of the Reviewer which greatly contributed to improve the quality of manuscript. The larje marjority of them were completely accepted. We have now completed all of the changes requested and the responses to Reviewer is provided bellow. In the text, the revised portions of the manuscript are highlight with red font.
"Line 18-19: Please use qualitative and quantitative word rather than variables."
"Over all abstract is poorly written and no has important information presented in abstract. Therefore please re-write it."
R – Abstract was completely revised and rewrote. We try add more information in order to attend the Reviewer suggestion.
"Introduction is well written just try to avoid short para."
R – The text was revised as suggested by Reviewer.
"Please cite the below related articles."
- Akhtar, K.; Wang, W.; Ren, G.; Khan, A.; Feng, Y.; Yang, G.; Wang, H., Integrated use of straw mulch with nitrogen fertilizer improves soil functionality and soybean production. Environment International 2019, 132, 105092
- Akhtar, K.; Wang, W.; Khan, A.; Ren, G.; Afridi, M. Z.; Feng, Y.; Yang, G., Wheat straw mulching offset soil moisture deficient for improving physiological and growth performance of summer-sown soybean. Agricultural Water Management 2019, 211, 16-25.
R – The citations were added in the introduction as suggested by Reviewer.
"Results and discussion are well written but i suggest to add sub-heading in the discussion section same as results."
R – We appreciate the suggestion, but we have decided not to add subheadings in the discussion as we have already elaborated the text in an integrative manner. Including subheading sections would result in subsections that could make the text repetitive.
"Conclusion is very short, explain and make it attractive."
R – The text was rewritten to promote a more attractive one.
"Statistical analysis portion is missing add it."
R – We add the sub-heading “Statistical analysis” as suggested.